# Diffusion-Based Co-Speech Gesture Generation Using Joint Text and Audio Representation

Anna Deichler
deichler@kth.se
KTH Royal Institute of Technology
Stockholm, Sweden

Shivam Mehta
smehta@kth.se
KTH Royal Institute of Technology
Stockholm, Sweden

Simon Alexanderson
simonal@kth.se
KTH Royal Institute of Technology
Stockholm, Sweden

Jonas Beskow
beskow@kth.se
KTH Royal Institute of Technology
Stockholm, Sweden

## ABSTRACT

This paper describes a system developed for the GENEA (Generation and Evaluation of Non-verbal Behaviour for Embodied Agents) Challenge 2023. Our solution builds on an existing diffusion-based motion synthesis model. We propose a contrastive speech and motion pretraining (CSMP) module, which learns a joint embedding for speech and gesture with the aim to learn a semantic coupling between these modalities. The output of the CSMP module is used as a conditioning signal in the diffusion-based gesture synthesis model in order to achieve semantically-aware co-speech gesture generation. Our entry achieved highest human-likeness and highest speech appropriateness rating among the submitted entries. This indicates that our system is a promising approach to achieve human-like co-speech gestures in agents that carry semantic meaning.

## KEYWORDS

gesture generation, motion synthesis, diffusion models, contrastive pre-training, semantic gestures

**ACM Reference Format:**
Anna Deichler, Shivam Mehta, Simon Alexanderson, and Jonas Beskow. 2023. Diffusion-Based Co-Speech Gesture Generation Using Joint Text and Audio Representation. In *INTERNATIONAL CONFERENCE ON MULTIMODAL INTERACTION (ICMI '23), October 09–13, 2023, Paris, France.* ACM, New York, NY, USA, 8 pages. https://doi.org/10.1145/3577190.3616117

## 1 INTRODUCTION

Human communication is inherently multimodal involving the integration of multiple verbal and non-verbal modalities to convey the information. These modalities work in synergy, collaborating to create a joint representation of the message the speaker intends to convey [29]. In addition to complementing verbal communication, these non-verbal gestures frequently serve as substitutes for words [9, 31]. The semantic meaning contribution of gestures is multi-faceted. Beat gestures primarily emphasize the verbally expressed content, serving to accentuate the spoken message. On the other hand, iconic and pointing gestures go beyond emphasizing content; they directly represent or indicate the referent being discussed. Deictic pointing gestures, often accompanying deictic words, play a crucial role in referential communication by providing vital contextual information for reference disambiguation, while iconic gestures serve to visually represent or symbolize the attributes, actions, or characteristics associated with the referent.

Co-speech gesture generation in robotics and avatars focuses on generating gestures that accompany and extend the verbal modality. However, the generation of audio-driven motion has posed a significant challenge. This difficulty arises from the fact that such motion can be accurately predicted by very strong probabilistic models, since gestures exhibit high individual variability, are inherently non-deterministic [2]. Recent advances in learning arbitrary probability distributions with diffusion models has offered a way to tackle this problem. These audio-driven gesture generation models have proven to be efficient in reproducing the high variability and expressivity of human gestures, however integrating semantic content into gesture generation by combining audio and text conditioning is another challenge.

Self-supervised pre-training methods have proven to be an efficient way to learn useful representations for downstream tasks, especially in case of limited labeled data. Multi-modal pre-training methods learn embedding spaces that encode useful relations of different data modalities. Contrastive Language-Image Pre-Training (CLIP) [32] is a contrastive multi-modal pre-training method that learns a joint representation of image and text data by contrasting positive and negative text-image pair examples in the latent space during training. This training approach encourage the model to capture the underlying relationship between the two modalities.

The problem of co-speech gesture generation involves multiple modalities, with a tight coupling between motion, text and audio. This work aims at combining the expressivity of diffusion based motion synthesis [2] with the multi-modal understanding of a CLIP-like latent embedding space that models the relations between motion, text and audio in co-speech gestures.

*ICMI '23, October 09–13, 2023, Paris, France*
© 2023 Copyright held by the owner/author(s).
ACM ISBN 979-8-4007-0055-2/23/10.
https://doi.org/10.1145/3577190.3616117

 Anna Deichler, Shivam Mehta, Simon Alexanderson, and Jonas Beskow

## 2 RELATED WORK

### 2.1 Co-speech gesture generation

The primary goal of co-speech gesture generation is to synthesise natural and contextually appropriate gestures. In the early stages of gesture generation research, various rule-based approaches were employed [5, 26, 27], where the generation of gestures was triggered by predefined rules that initiated the playback of pre-recorded gestures. In recent years, this field has been dominated by the use of data-driven deep learning based modelling methodologies [31].

Early works on deep learning-based gesture synthesis treated it as a regression problem and utilised recurrent [14, 36] and convolutional [21] neural networks to model the generation process. Treating gesture synthesis as a regression problem leads to the problem of under-articulated and over-smoothened gestures because of averaging over all the possible outcomes for an input signal. To address the challenge of under-articulated and over-smoothened synthesis researchers employed various probabilistic modelling techniques such as VAEs [12], VQ-VAEs [43], Normalising Flows [1] or adversarial techniques like GANs [41, 42]. These methodologies aim to enhance the realism and expressiveness of the generated gestures by learning a distribution over the entire utterances and sampling different realisations from it or learning powerful transformations from a simple distribution, usually a Gaussian distribution, to the output motion distribution.

Diffusion models [15, 34, 35] have emerged as a notable and contemporary probabilistic generative modelling methodology. These models have shown promise in capturing complex data distributions and have gained attention in various fields, including gesture generation [2, 3, 30, 45]. Inspired by these works our system uses Denoising Diffusion Probabilistic Modelling (DDPM) [15] formulation with self-supervised representations to synthesise gestures conditioned on the input audio.

### 2.2 Semantic gesture generation

In order to generate contextually appropriate gestures in agents, it is crucial to take into account gesture semantics. Semantic gestures have a symbolic representational quality and contribute to the overall meaning in communication. The generation of semantic gestures is highly reliant on which input modalities are taken into account in the modeling process [31].

Audio driven generation can reproduce the coupling between gesture kinematics and the intonation, stress and rhythm present in the audio signal. These systems are good at modeling beat gestures, which can help highlight important points or add emphasis to certain words or phrases [28],[1],[2]. However, in order to generate representational gestures (e.g., iconic, deictic pointing), additional input modalities are needed. Text-based conditioning is essential to model the relation between semantic and kinematic spaces in order to generate iconic gestures [44],[22], while the generation of deictic pointing gestures needs referential target information [10]. In this work we develop a novel approach to jointly model audio and text conditioning in gesture generation through a contrastive self-supervised learning approach in order to extend the existing audio conditioned system with semantic capabilities.

### 2.3 Using language based pre-training approaches in motion generation

Recent works approaches have leveraged different pre-training approaches to learn the semantic coupling between text and motion spaces. [46] uses a GPT-like module to generate code indices based on text embeddings which are utilized by a VQ-VAE module in motion generation, while [17] proposes MotionGPT, which performs language modeling on both motion and text in a unified manner, treating human motion as a specific language. Previous work has also leveraged CLIP's multimodal understanding to generate meaningful motion. [37] develops an auto-encoder based motion generation model, which learns a motion embedding space aligned with CLIP's latent space, which allows for the generation of expressive and versatile text-based motions. [38] uses CLIP latents as conditioning information in diffusion based human motion generation. Similarly, [8] conditions on CLIP latents, but combines latent space based and diffusion based motion generation. Most similar to our work is [3], which learns a gesture-text joint embedding using contrastive learning and a CLIP based style encoding module in a diffusion based gesture synthesis model.

## 3 METHOD

### 3.1 Self-supervised representations of text and audio

We employ pre-trained self-supervised representations for text and audio for both the main agent and the interlocutor. Data2vec [4] which is a framework for self-supervised representation learning on data of different modalities (text, audio and images), for which pre-trained models are available [1]. Data2vec leverages transformer architecture in a self-distillation setup to achieve contextual text embedding, predicting latent representations of the full input data based on a masked view of the input.

For audio, we use the `data2vec-audio-base-960h` model, which takes one-channel 16 Khz audio as input. As output we use the last hidden layer, which gives us a sequence of 768-dimensional embedding vectors at a rate of 50 Hz. The output is then converted to 30 Hz using polyphase resampling (`scipy.signal.resample_poly`) in order to match the frame rate of the motion data.

For text, we use the `data2vec-text-base` model. Input to the model is a sequence of byte-pair encoded text tokens. Just as for the audio, we use the last hidden layer of the data2vec model to obtain a 768-dimensional vector for each input token. We use the word timed transcriptions provided in the dataset (see [23]) to maintain a start and end time for each token, then we replicate the output vector at a rate of 30 Hz for the duration of the token, The result is a text-embedding sequence that is aligned with, and of the same length as, the audio and motion data sequences.

### 3.2 Join representation with Contrastive Speech and Motion Pretraining (CSMP)

Contrastive pre-training can effectively capture the semantic relationships between two modalities but usually, it requires a larger batch size and larger dataset to learn efficient joint representations [7] which can be challenging especially in this case because of

---

[1]huggingface.co

dataset-specific properties [23] such as the presence of an interlocutor and the skeletal nodes of the characters. In such a case having representations which already capture semantic information can be used as the inputs to the CLIP module. Therefore, we devise a variation of CLIP and call it Contrastive Speech and Motion Pretraining (CSMP).

In CSMP, we propose several modifications to the original CLIP architecture within the context of multimodal understanding namely:

(1) We replace the vision transformer present in the original CLIP architecture with a regular transformer architecture, which effectively eliminates the patching process typically employed for 2-D image analysis. This modification is motivated by the nature of text and audio.
(2) The input to this modified transformer is derived from concatenated representations of the output of the pretrained data2vec module for text and audio as described in section 3.1 instead of raw tokens for original CLIP.
(3) For the text encoder in CLIP, we modify the input from discrete text tokens to continuous motion vectors thus eliminating the need for an embedding layer. This alteration is intended to mimic the semantic information contained in the text and audio representations to the motion representation in the joint space of CSMP's representations.
(4) Since the original clip takes discrete and tokenized text as an input it had a context length of 77 this, in the case of modalities like the output of data2vec and motion which is continuous in nature can be insufficient to capture longer-term dependencies. In order to overcome and increase the encoder's field of view we increased the context length to 500 timesteps.

The final architecture of the CSMP module is described in Fig. 1

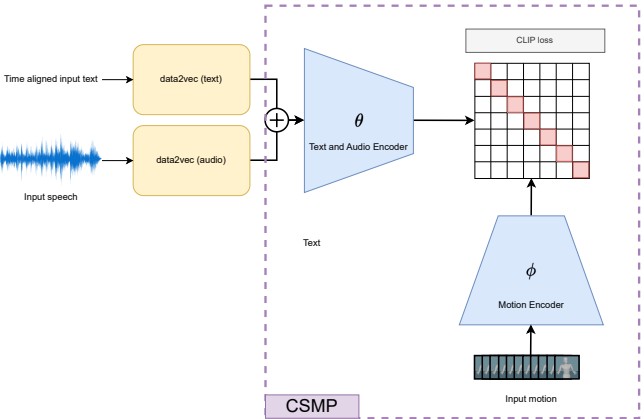

**Figure 1: Architecture of Contrastive Speech Motion Pretraining (CSMP) module.**

In order to train such an architecture with CLIP loss, we chunked each input $X_i = [x_1, \cdots, x_T]$ in a sliding window manner with a window length of 500 and a hop length of 250 and formed multiple splits for each utterance.

$$X_i = [[x_1, \cdots, x_{500}], [x_{250}, \cdots, x_{750}], \cdots, [x_{T-500}, \cdots, x_T]]$$

We hypothesise that this helped in the generalisation despite a fixed context size because the positional encoding could see the data at a specific timestep $x_t$ in different relative positions while training. The source code is available on GitHub in GestCLIP branch[2].

## 3.3 DDPM for motion synthesis

Diffusion models are a recent class of generative models that have become popular due to their expressivity and flexible conditioning. Diffusion models are based on the idea that complex data distributions can be learned by iteratively transforming a simple known distribution, such as a Gaussian distribution, through a series of diffusion steps. Unlike VAEs, which incorporate latent variable modeling, diffusion models directly model the data distribution without explicitly introducing latent variables. Diffusion models consist of a forward process and a reverse (denoising) process. The forward process defines a Markov chain of $N$ diffusion steps to gradually add noise to samples from the data distribution $x_0 \sim q(z)$. The noise steps are assumed to be fixed, zero-mean Gaussian distributions, without learnable parameters, $q(x_n|x_{n-1}) = \mathcal{N}(x_n; \sqrt{1 - \beta_n}x_{n-1}, \beta_n \mathbf{I})$, where $\mathcal{N}$ denotes the multivariate Gaussian density function evaluated at $x_n$ and $\{\beta_n\}_{n=1}^N$ is the noise schedule. In the reverse process the model learns to reverse the forward process so that the model is able to construct desired data samples from the noise. If $\beta_n$ is small enough, the reverse step $p(x_{n-1}|x_n)$ is also Gaussian and a neural network is used to approximate the parameters of the distribution $p_\theta(x_{n-1}|x_n) = \mathcal{N}(x_{n-1}; \mu_\theta(x_n, n), \Sigma_\theta(x_n, n))$.

The Denoising Diffusion Probabilistic Model (DDPM) [15] simplifies the objective of diffusion model and establishes a connection to score matching, which is a technique used for estimating the gradients of the probability distribution of data. These gradients are then used to generate samples via Langevin dynamics, which is a stochastic process that simulates the motion of particles in a fluid. In DDPM the score-matching objective is reformulated as noise predicting objective, $\mathcal{L} = \mathbb{E}_{x_0,n,\epsilon}[\kappa_n \|\epsilon - \epsilon_\theta(x_n, n)\|_2^2]$, where $\epsilon_\theta$ is a neural network intended to predict the noise $\epsilon$ that was added to $x_0$ and $\kappa_n$ are weights.

Conditional generation in diffusion models can be achieved via classifier-guided or classifier-free models. In classifier guided diffusion models the gradients of a separately trained classifier $f_\phi(y|x_n)$ is used to guide the diffusion process $\nabla_\mathbf{x} f_\phi(y|x_n)$[11]. Classifier-free diffusion models combine conditional and unconditional diffusion in order to guide the diffusion. In the above formulation this means that a conditional network $\epsilon_\theta(x_n, n, c)$, with conditioning input $c$ is trained, where the conditioning information is randomly discarded during training, so that in the reverse diffusion process conditional generation can be achieved by the combination of the input conditioned and unconditioned model $\bar{\epsilon}_\theta(x_n, n, c) = \epsilon_\theta(x_n, n, c) + \gamma(\epsilon_\theta(x_n, n, c) - \epsilon_\theta(x_n, n))$ [16]. Denoising diffusion based conditional generation has been applied in various domains. In [33], the CLIP embedding based conditioning input is randomly set to zero in order to achieve high quality image synthesis. DiffWave [20] is a denoising diffusion based model for waveform generation, which uses mel spectrograms and speaker ID as conditioning information. The Listen-Denoise-Act (LDA) model

---

[2]https://github.com/shivammehta25/CLIP/tree/GestCLIP

[2] builds on the DiffWave model and uses mel spectogram information for human motion synthesis. Audio based conditional human motion synthesis, such as dancing and co-speech gesture generation have been a challenge in machine learning, due to the ambiguity and high versatility required for good performance in these tasks. The denoising diffusion based LDA model have proven to be a powerful model to generate versatile and expressive motion in the fields of dance and co-speech gesture generation. In our work we use the residual deonising network of LDA with a conditioning from the CSMP module for semantically-aware co-speech gesture generation.

The LDA model follows DiffWave in parameterising the denoising network $\epsilon_\theta$, but replaces the dilated convolutions in the stacked residual blocks with a stack of Transformers [39] or Conformers [13] in order to capture and integrate information over long time scales. In our experiments we use a stack of 3 translation-invariant transformers [40] in each of the 15 residual blocks. The model learns a distribution of the form $p(x_{1:T}|a_{1:T})$, where $a_{1:T}$ is the acoustic conditioning and $x_{1:T} = x_{1:T,0}$ is the output of the diffusion process and $x_t$ is a representation of the pose at time step $t$ in the motion sequence. In our case, the mel spectrogram based acoustic conditioning of LDA is replaced with the joint audio and text based output of the CSMP module, where the outputs for interlocutor and the main agent data are concatenated into a conditioning signal of dimension $c_t \in \mathbb{R}^{1024}$. This is the conditioning input in the classifier-free diffusion guidance formulation. The outputs of the model are the same in LDA, poses of skeletal joint rotations parametrised using an exponential map representation relative to a T-pose, similarly as in [1].

## 4 DATA PREPARATION

The challenge dataset is a processed version of the Talking With Hands dataset[25]. The original dataset is one of the largest conversational dataset of motion and voice, incorporating 50 hours of dyadic interactions, with audio, text and motion modalities. We only used the data provided by the challenge for gesture synthesis.

### 4.1 Audio DC-removal and muting of cross-talk

We found that the audio data contained a number of loud transient clicking noises. On inspection, it was found that they were due to a significant DC-offset, in combination with the fact that certain sections of the audio signal had been zeroed out, as part of an anonymization process. This was easily rectified by subtracting the mean from all non-zeroed out portions.

Additionally, the data contained a non-negligible amount of cross-talk between the two speakers in the recording. We used the time stamps from the time-aligned text transcriptions to mute all audio falling outside of the intervals marked as speech in the transcription for each speaker. We used a 200 ms ramp function for the muting to avoid introducing transients.

### 4.2 Motion capture data cleaning

We also noticed that some of the motion capture data contained errors such as joints suddenly popping to unnatural poses. These errors were predominantly confined to the wrist joints, but also occurred at the hips. As such problems has an impact model training,

and we even found our model reproducing them in synthesis, we performed some data cleanup. We transformed the data to joint positions and detected discontinuities in the wrist speeds using a Hampel filter. This was followed by a manual check of the affected files. In the end, 17 files were removed from the training set.

## 5 SYSTEM OVERVIEW

Schematic view of the final system can be seen in Figure 2. The system was trained on a NVIDIA GeForce RTX 3090 for $387.4k$ steps and achieved 0.013 loss on the training and 0.019 loss on the validation set. No post-processing was applied on the generated output motions.

## 6 EVALUATION

The evaluation of the generated motions was carried out by the GENEA Challenge organisers, details about the evaluation interface and experiment setups can be found in the evaluation paper [24]. The generated co-speech gesture were evaluated in three separate perceptual studies: human-likeness, appropriateness to the agent's speech and appropriateness to the interlocutor's motion and speech. The evaluation included two baseline conditions and the natural motion taken from the motion-capture recordings. The monadic baseline ('BM') was generated with [6] which uses information from the main-agent for gesture generation, while the dyadic baseline ('BD') is an adapted version of the former, which also includes information from the interlocutor in the conversation. The study participants were recruited through a crowd-sourcing platform from English-speaking countries and each study incorporated attention checks. Our system, labeled as 'SG' achieved top performance in the studies of human-likeness and speech appropriateness based on the generated motions submitted. However, it ranked among the lowest in terms of interlocutor appropriateness.

*6.0.1 Human-likeness evaluation.* The aim of this study was to evaluate whether the generated motion of the virtual character looks like the motion of a real human. No audio was used in order to disentangle the human-likeness evaluation from the speech appropriateness. The evaluation was based on the HEMVIP methodology [19], where multiple different motion samples are presented in parallel and the participant is asked to rate each sample. Participant could give their ratings on a scale from 0 (worst) to 100 (best). Results for the evaluation are shown on Figure 3. Our system, denoted as 'SG', achieved best performance from the entries, with mean rating of $65.6 \pm 1.4$. Figure 4 also shows that this results is significantly better than all of the entries, except 'SF'. Interestingly, the human-likeness score is very close to mean rating of the natural condition, which was $68.4 \pm 1.4$ as seen on Table 1. This indicates that our system can generate co-speech gestures which resembles the motion of real humans.

*6.0.2 Appropriateness to speech.* The aim of this study was to evaluate whether the motion if the virtual character is appropriate for the given speech, controlling the overall human-likeness of the motion. The participants were presented with a pair of matched and mismatched videos from the same condition in order to disentangle this study from the motion quality evaluation. Five response options were given for indicating preference over the 2 videos and

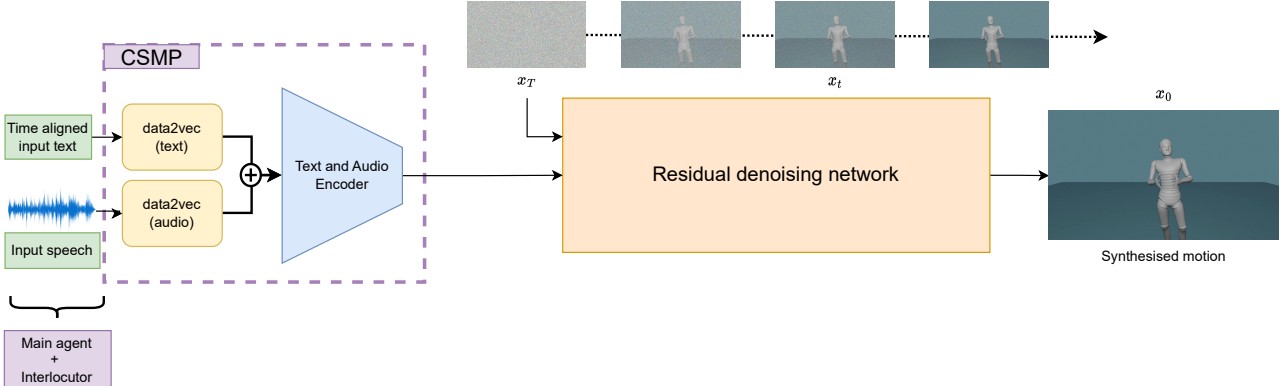

Figure 2: Architecture of the motion synthesis module

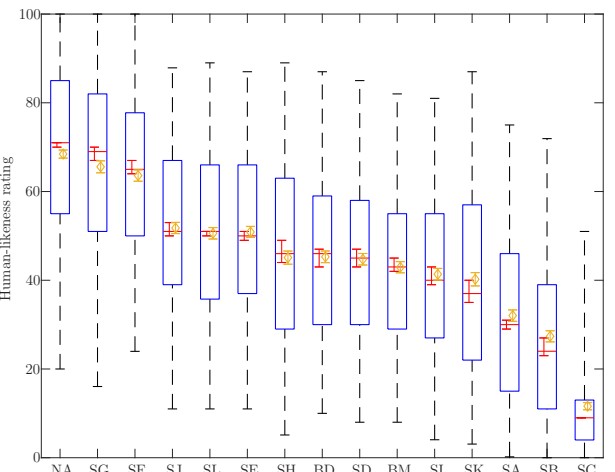

Figure 3: Box plot visualising the ratings distribution in the human-likeness study. Red bars are the median ratings (each with a 0.05 confidence interval); yellow diamonds are mean ratings (also with a 0.05 confidence interval). Box edges are at 25 and 75 percentiles, while whiskers cover 95% of all ratings for each condition. Conditions are ordered by descending sample median rating.

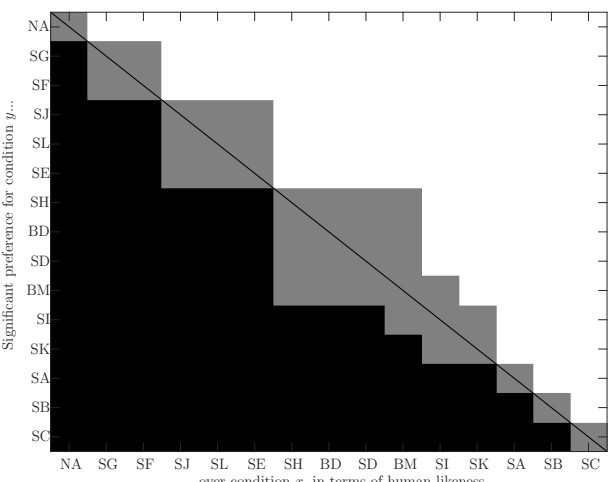

Figure 4: Significance of pairwise differences between conditions. White means the condition listed on the $y$-axis achieved an MAS significantly above the condition on the $x$-axis, black means the opposite ($y$ scored below $x$), and grey means no statistically significant difference at level $\alpha = 0.05$ after correction for the false discovery rate. Conditions use the same order as the corresponding subfigure in Figure 3

the responses were converted to integer values in the range of $[-2, 2]$. Our system achieved a MAS score of $0.39 \pm 0.07$ at the level of $\alpha = 0.05$ and the matched motion was preferred over the mismatched in 61.8% of the evaluations. With these results it ranked highest amongst the generated motions. Figure 5 visualizes the significant differences between conditions and shows that our system, denoted by 'SG', was significantly more appropriate to speech than all of the entries of generated motions. Comparison to other entries can be found in Table 1.

*6.0.3 Appropriateness to interlocutor.* The aim of this study was to evaluate whether the motion of the virtual character is appropriate for the given interlocutor behavior (speech and motion). In order to evaluate the mismatched condition, synthetic interactions were created, where the main agent was the same, but the interlocutor behavior was replaced with one from another interaction. Our system achieved a MAS score of $-0.09 \pm 0.08$ at the level of $\alpha = 0.05$ and the matched motion was preferred over the mismatched in 46.7% of the evaluations. With these results it ranked among the lowest. Figure 6 visualizes the significant differences between conditions and shows that our system, denoted by 'SG', was significantly less appropriate to interlocutor than all half of the entries of generated motions and there was no significant difference to the other half. Comparison to other entries can be found in Table 1.

The MP4-format video stimuli used in the user studies can be accessed through the following link: https://zenodo.org/record/8211449. As before, our system is denoted as 'SG'.

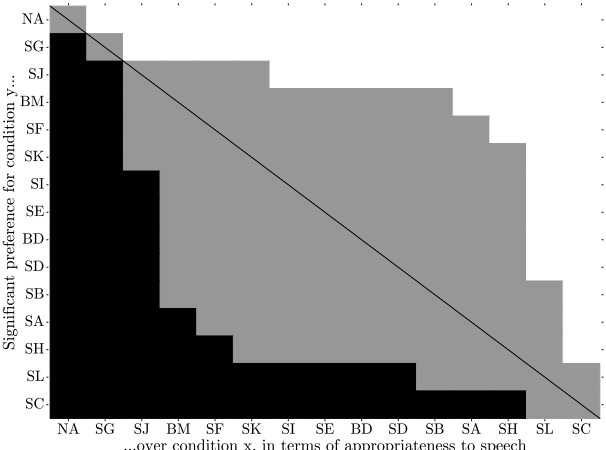

Figure 5: Appropriateness for agent speech

Figure 6: Appropriateness for the interlocutor

Figure 7: Significant differences between conditions in the two appropriateness studies. White means the condition listed on the $y$-axis achieved an MAS significantly above the condition on the $x$-axis, black means the opposite ($y$ scored below $x$), and grey means no statistically significant difference at level $\alpha$ = 0.05 after correction for the false discovery rate.

Table 1: Summary of results for subjective evaluation studies with confidence intervals for the mean appropriateness score (MAS) at the level $\alpha$ = 0.05. "Pref. matched" identifies how often test-takers preferred matched motion in terms of appropriateness, after splitting ties equally.

| | Human-likeness | | Speech appropriateness | | | Interlocutor appropriateness | | |
|---|---|---|---|---|---|---|---|---|
| Condition | Median | Mean | Condition | MAS | Pref.M. | Condition | MAS | Pref.M. |
| NA | 71 ∈ [70, 71] | 68.4±1.0 | NA | 0.81±0.06 | 73.6% | NA | 0.63±0.08 | 67.9% |
| SG | 69 ∈ [67, 70] | 65.6±1.4 | SG | 0.39±0.07 | 61.8% | SA | 0.09±0.06 | 53.5% |
| SF | 65 ∈ [64, 67] | 63.6±1.3 | SJ | 0.27±0.06 | 58.4% | BD | 0.07±0.06 | 53.0% |
| SJ | 51 ∈ [50, 53] | 51.8±1.3 | BM | 0.20±0.05 | 56.6% | SB | 0.07±0.08 | 51.8% |
| SL | 51 ∈ [50, 51] | 50.6±1.3 | SF | 0.20±0.06 | 55.8% | SL | 0.07±0.06 | 53.4% |
| SE | 50 ∈ [49, 51] | 50.9±1.3 | SK | 0.18±0.06 | 55.6% | SE | 0.05±0.07 | 51.8% |
| SH | 46 ∈ [44, 49] | 45.1±1.5 | SI | 0.16±0.06 | 55.5% | SF | 0.04±0.06 | 50.9% |
| BD | 46 ∈ [43, 47] | 45.3±1.4 | SE | 0.16±0.05 | 54.9% | SI | 0.04±0.08 | 50.9% |
| SD | 45 ∈ [43, 47] | 44.7±1.3 | BD | 0.14±0.06 | 54.8% | SD | 0.02±0.07 | 52.2% |
| BM | 43 ∈ [42, 45] | 42.9±1.3 | SD | 0.14±0.06 | 55.0% | BM | -0.01±0.06 | 49.9% |
| SI | 40 ∈ [39, 43] | 41.4±1.4 | SB | 0.13±0.06 | 55.0% | SJ | -0.03±0.05 | 49.1% |
| SK | 37 ∈ [35, 40] | 40.2±1.5 | SA | 0.11±0.06 | 53.6% | SC | -0.03±0.05 | 49.1% |
| SA | 30 ∈ [29, 31] | 32.0±1.3 | SH | 0.09±0.07 | 52.9% | SK | -0.06±0.05 | 47.4% |
| SB | 24 ∈ [23, 27] | 27.4±1.3 | SL | 0.05±0.05 | 51.7% | SG | -0.09±0.08 | 46.7% |
| SC | 9 ∈ [9, 9] | 11.6±0.9 | SC | -0.02±0.04 | 49.1% | SH | -0.21±0.05 | 44.0% |

## 7 DISCUSSION

The subjective evaluation results have shown that our system is capable of generating of co-speech gestures that are human-like and speech appropriate. The high performance on the speech appropriateness shows that the current system is a promising approach to achieve semantically-aware co-speech gesture generation in virtual agents.

Our system was top-ranked in the human-likeness and appropriateness for agent speech evaluations, while receiving one of the lowest scores in the appropriateness to interlocutor evaluation. This

might seem a bit counter intuitive, given that we indeed trained the system to listen to the interlocutor. We believe that there are multiple factors at play here and will outline them below. First, out system was trained to take in speech information of the interlocutor as input (in the form of CSMP embeddings), but we chose to not include interlocutor motion as one of the inputs, due to time constraints. Feeding interlocutor motion as input might have rendered a system capable of mirroring/mimicry, similar to [18] which could have resulted in a higher rating. Secondly, we would like to discuss another possible explanation, which stems from the nature of the

data and how the evaluation was carried out. In the appropriateness evaluations, each system was compared against itself, and the objective was to see to what degree raters could distinguish motion that matched the context from mis-matched motion. As mentioned in section 4.1, there was a certain amount of cross-talk present in the data, i.e. the interlocutor audio was present in the main agent's channel and vice versa. We took extra measures to eliminate such cross-talk, because not doing so would have resulted in the agent performing co-speech gestures also while listening, based on the cross-talk from the interlocutor. Inspecting the evaluation stimuli based on the output from the different systems in the challenge, it is clear that this seems to happen in certain systems. We can further speculate that such an agent might in fact score favourably in the match/mismatch paradigm, because the gestures would indeed be interlocutor aware. Future work on improving the interlocutor appropriateness could involve conditioning on interlocutor motion, as mentioned above, or training a separate model for listening behavior.

Additional evaluations on the semantic gesture generation capabilities of the model could be of interest for future work. In theory, our model is capable of capturing the semantic relations between speech and gesture spaces through the CSMP model. However, the current subjective evaluation is a bit limited in measuring the semantic gesture generation capabilities of the model, as it is difficult to disentangle from other aspects, such as speech-gesture synchrony. Objective evaluation metrics for semantic appropriateness could be helpful in quantifying and improving our system in this regard.

## 8 CONCLUSIONS

In this paper we described our entry system to the GENEA Challenge 2023. We presented a system, which builds on an existing diffusion based motion synthesis model and proposed a conditioning signal, which utilizes audio, text and motion data. For this we proposed a CLIP-like contrastive pre-training module, contrastive speech and motion pretraining (CSMP) in order to capture the underlying relations between speech and motion. Our system achieved top performance in human-likeness and speech appropriateness amongst the submitted entries, which proves that our system is a promising approach to generate human-like co-speech gestures in agents. Our system ranked relatively low in interlocutor approrpiateness, which is a focus in future work for improvement. Human-like, semantic and interlocutor appropriate co-speech gesture generation in virtual agents is still an open problem. Our systems high performance in the subjective evaluations is encouraging and indicates that our submitted model is a promising way to achieve these goals.

## ACKNOWLEDGMENTS

This work was partially supported by the Advanced Adaptive Intelligent Agents project (Digital Futures), the Wallenberg AI, Autonomous Systems and Software Program (WASP) funded by the Knut and Alice Wallenberg Foundation and by grant no. 20023495 (Development of behavior-oriented HRI AI technology for long-term interaction between service robots and users) funded by the Korean Ministry of Trade, Industry and Energy (MOTIE).

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
