# OpenReview forum: "Diffusion-based co-speech gesture generation using joint text and audio representation"
_ACM.org/ICMI/2023/Workshop/GENEA_Challenge — GENEA Challenge 2023 Mainproceeding_

### Official Review · Reviewer_bJQw · 2023-07-30
**This paper introduces a diffusion-based co-speech gesture synthesis system, effectively using self-supervised multimodal features. I choose to recommend its acceptance to the ICMI workshop.**

**Rating:** 8
**Confidence:** 5

**Review:**

This paper proposes a diffusion-based co-speech gesture synthesis system, which successfully explores the use of self-supervised pre-trained joint multimodal input features. How to efficiently mine the multimodal relations between speech and gestures via the CLIP-like pre-training strategy, I believe, is the right direction. The challenge results achieved by this paper strongly support this claim.

Main Pros:
* A diffusion-based model achieving high motion quality;
* Employment of contrastive speech motion pre-training, enhancing the semantic awareness of the system.

Typos & Questions:
* Line 10-11: Our "solutions" -> "solution";
* Offering some visualization results is encouraged.

To sum up, I think this paper proposes an efficient framework to solve the weak semantics problem in the field of deep-based gesture generation. I would like to recommend that this work be accepted to the ICMI workshop.

---

### Official Review · Reviewer_D2gH · 2023-07-31
**Contrastive pretrained speech feature extractor with good results.**

**Rating:** 7
**Confidence:** 5

**Review:**

This paper present a novel method of speech feature extraction for co-speech gesture generation task. It uses contrastive learning to pretrain a speech encoder to extract the most relevant information with the motion. The paper is clearly written and in detail. An explanation on the discrepancy between the evaluation results is also provided.

The idea of utilizing pretrained models, i.e., data2vec, as backbone before the encoders in the contrastive learning is interesting. The authors claim that it maybe a way to overcome the lack of large amount of speech-gesture data and I agree with it although more experiments are necessary. It is great to see attempts on using pretrained audio and text models as they may become informative guidance of building a more powerful gesture generation systems.

One minor point is that the authors do not have explanations about why their approach works better. Although ablation study could be difficult in the current setting, qualitative results can provide insights as well.

One question regarding the generation.
The author is using classifier-free guidance in the generation. However, a CLIP that accepts noised version of motion can be trained to apply classifier guidance, which has been proposed in conditional image generation previously [1]. I am wondering why the authors chose classifier-free guidance over classifier guidance.

[1] GLIDE: Towards Photorealistic Image Generation and Editing with Text-Guided Diffusion Models

---

### Decision · Program_Chairs · 2023-08-04

**Decision:**

Accept (Main proceeding)

**Comment:**

As both reviewers recommended an acceptance and this meta-reviewer agreed on that, the chairs have decided to accept this paper to the Main ICMI Proceedings. The contrastive pre-training is interesting and the results showed the superiority of the proposed method. Please prepare the camera-ready version based on the reviews.

Also, please make sure to update reference 36 which is the main challenge paper. The bibtex was in the guideline document that we provided earlier.